# Effect of the Use of Home and Community Care Services on the Multidimensional Health of Older Adults

**DOI:** 10.3390/ijerph192215402

**Published:** 2022-11-21

**Authors:** Qun Wang, Kunyi Fan, Peng Li

**Affiliations:** Faculty of Humanities and Social Sciences, Dalian University of Technology, Dalian 116024, China

**Keywords:** home and community care, physical health, mental health, social adaptation

## Abstract

Home and community care is an important way to actively respond to population aging and to promote healthy aging. This study aims to estimate the effect of using home and community care services on the multidimensional health of older adults. We used data from the China Health and Retirement Longitudinal Study conducted in 2018 and relied mainly on the propensity score matching method for data analysis. The results showed that using home and community care increased the probability of maintaining and improving physical health by 2.9%, decreased the score of depression by 0.471, and improved the score of cognitive function by 0.704. Using home and community care also increased the probability of actively participating in life by 4.1% and elevated the score of life satisfaction by 0.088. The heterogeneity analysis showed that the use of home and community care had a significant effect on promoting all health indicators in rural older adults and a more obvious promoting effect on the social adaptation of urban older adults. Using home and community care significantly promoted the multidimensional health of people aged 60 to 79 years but had no impact among people aged ≥ 80 years. The use of home and community care significantly improved all health indicators in non-disabled older adults. Whereas, it only improved the levels of cognitive function and life satisfaction in disabled older people. Using this form of care significantly improved all health indicators in those with low socio-economic status, but it only had a partial positive effect on the multidimensional health of those with high socio-economic status. Our results are of importance to the government as they may be used to further improve the quality of home and community care services for the targeted older population.

## 1. Introduction

As the aging process accelerates, care for older adults has become an increasingly important social issue to be solved globally. Currently, there are two main types of long-term care services available for older adults: institutional care, and home and community care. Compared with institutional care, the advantages of home and community care services are that they can provide diversified services for older people at a relatively reasonable price and meet their emotional needs not to leave home [1,2]. Such advantages make home and community care services an important way to actively respond to population aging and promote healthy aging. Therefore, during the last decade, low- and middle-income countries (LMICs), especially China, have started to develop home and community care services for older adults [3,4]. As of 2020, China had 264.02 million people aged 60 and above, accounting for 18.7% of the total population [5]. The Chinese government aims to build a “9073” long-term care service system, i.e., about 90% of older adults rely on home care, about 7% on community care, and about 3% on institutional care. Considering the huge demand for home and community care, such services in China will develop very quickly in the future [6]. Thus, in these times of increasingly vigorous developments in home and community care services in China, it is of great significance to systematically evaluate the impact of using home and community care services on the multidimensional health of older adults to meet their urgent needs for long-term care services with high quality.

As the WHO stated, some changes in the health of older adults are hereditary, but most are caused by the natural and social environment in which people live—including families, neighborhoods, and communities [7]. Grossman proposed that health-related service input will increase the health stock of individuals through a specific production function [8]. Empirically, the evaluation of home and community care services has also attracted researchers’ attention. Some studies compared quality, quality of life, satisfaction levels, etc. between home and community care and other forms of long-term care, such as institutional care [9,10]. Other studies have evaluated the effectiveness and quality of care homes, and community care itself, but the majority were only based on a small sample of users [11,12]. Few studies have evaluated the relationships between home and community care and life satisfaction, loneliness, perceived life stress, and cognitive function of a large sample containing users and non-users. However, they either used simple logistic regression without accounting for endogeneity problems [13] or focused on the availability of community care on one dimension of health, i.e., without taking the perspective of service utilization [14]. In addition, most evidence related to home and community care was based in high-income countries. Little existed in LMICs. This study provides empirical evidence on the relationship between using home and community care and the multidimensional health of older adults in China.

## 2. Materials and Methods

### 2.1. Data

We used data from the China Health and Retirement Longitudinal Study (CHARLS) in 2018, which representatively collected social, demographic, economic, health, retirement, etc. data among the population in China aged ≥ 45 years. We only kept the data of those who were aged ≥ 60 years in 2018. Ultimately, we included 9692 respondents in the analysis. The data from the CHARLS was obtained via an online application.

### 2.2. Variables and Their Measurement

The outcome variable in this study was the health status of older adults and it had three dimensions: physical health, mental health, and social adaptation [15]. The physical health dimension was centered on the change in physical health. We used the question: “Compared with your health when we talked with you last time, would you say that your health is better now, about the same, or worse?” If the answer to this question was “better” or “about the same”, the variable was defined as 1, which represents maintained or improved physical health, otherwise, the variable was defined as 0. The mental health dimension selected indicators of depression and cognitive function. Depression was scored according to the answers to the ten questions in the Center for Epidemiologic Studies Depression Scale. For the eight negative emotion questions, we scored the answers “rarely” as 0, “some days” as 1, “occasionally” as 2, and “most of the time” as 3. The scores for the two positive emotion questions were the opposite (i.e., “rarely” was 3, and “most of the time” was 0) [16]. The measurement of cognitive function referred to the simple mental state examination scale, which included time orientation, place orientation, calculation, memory, reading, and writing [17]. The higher the score, the higher the cognitive function. The measurement of social adaptation was based on the conceptual framework proposed by Larson, which included objective social networks and social relations, in addition to subjective satisfaction with interpersonal relationships and social roles [18]. We measured social adaptation from objective and subjective levels. The objective level utilized questions relating to active participation in life [19]. If the respondent participated in at least one of these three activities in the past month: “providing help to relatives, friends or neighbors who do not live with you”, “participating in community organization activities”, and “voluntary activities or charitable activities”, the objective social adaptation was defined as 1, otherwise it was defined as 0. To measure the subjective social adaptation, we used the question, “Are you satisfied with your life in general”. The answer to this question had a 5-scale metric. We scored “extremely satisfied” as 5 and “not at all satisfied” as 1.

The explanatory variable was whether older adults had used home and community care services, including regular physical examinations, onsite visits, family beds, dining tables for older adults, services from daycare centers, etc. If older adults had used at least one, the respondent was defined as having used home and community care, otherwise it was defined as 0.

As for covariates, we selected two categories: individual characteristics and family characteristics. Only three covariates, i.e., ADLs (Activities of Daily Living), IADLs (Instrumental Activities of Daily Living), and socioeconomic status (SES), needed explanations. ADLs included six activities: dressing, bathing, eating, getting up, using the toilet, and controlling urine. IADLs included six tasks: housework, cooking, shopping, making phone calls, taking medicine, and managing money. Each question that related to ADLs and IADLs had four answer categories. We scored “no difficulty” as 0, “difficult but still can be completed” as 1, “difficult, need help” as 2, and “unable to complete” as 3. In the main analysis, ADLs and IADLs were used as continuous variables (Table 1). In the heterogeneity analysis, respondents who replied, “difficult but still can be completed”, “difficult, need help”, or “unable to complete” to one or more of the ADLs or IADLs were defined as disabled older adults [20]. We used the annual per capita household consumption as the SES indicator. In the main analysis, SES was used as a continuous variable, while SES was used as a binary variable in the heterogeneity analysis.

### 2.3. Statistical Methods

To overcome endogeneity problems, we mainly used the propensity score matching (PSM) method to estimate the average treatment effect (ATT) of the use of home and community care on the health of older adults. This method matches the characteristics of home and community care users (the treatment group) with non-users (the control group), so the difference between users and non-users was observationally equivalent except for the explanatory variable: whether they used home and community care or not [21]. We included the selected covariates as the matching variables and used the logit model to estimate propensity scores. We utilized kernel matching, k-nearest neighbor matching (k = 4), and radius matching to match the treatment and control groups. We carried out the balance test and common support test to ensure the quality of the matching. Based on the matching results, we estimated the ATT of using home and community care services on the health of older adults.

## 3. Results

### 3.1. Sample Characteristics

Among the 9692 respondents, 1957 (20.20%) had used home and community care services and 7735 (79.80%) had not. Of the total, 4594 (47.40%) respondents maintained or improved their health, and 5098 (52.60%) deteriorated their health. The average score for depression, cognitive function, and life satisfaction was 8.596, 21.176, and 3.280, respectively. Of the entire sample, 8481 (87.50%) failed to actively participate in life.

### 3.2. Estimates of Propensity Scores Using a Logit Model

We used the logit model to estimate propensity scores, which is the first step in applying the PSM method. Table 2 shows that age, being an empty nester, the number of chronic diseases, IADLs, smoking, drinking, the number of children, and health insurance significantly affected the use of home and community care services among older adults (*p* < 0.05).

### 3.3. The Quality of Matching

Figure 1 and Table 3 show the results of the balance test estimated by nuclear matching. Figure 1 illustrates that the standard deviation of each variable reduced after matching, albeit by less than 10%. Table 3 shows that all the variables, except age, did not reject the null assumption that no systematic difference existed between the treatment and control groups, which effectively solved the endogenous bias caused by the sample selection bias.

Figure 2 demonstrates the results of the common support test estimated by nuclear matching. Most of the observations in the treatment and control groups related to common support, and only two observations in the treatment group, and 13 observations in the control group were excluded because they were off support. Among the remaining sample, 7722 were in the control group, and 1955 were in the treatment group (and other matching methods excluded similar observations).

### 3.4. The Effect of Using Home and Community Care on the Health of Older People

Table 4 shows the PSM results based on the three different matching methods. We found that the use of home and community care services significantly increased the multidimensional health of older adults. Specifically, based on the nuclear matching method, the use of home and community care services significantly increased the possibility of maintaining or improving physical health by 2.9%, decreased the score of depression by 0.479, and increased the score of cognitive function by 0.694. Furthermore, the use of home and community care services significantly increased the probability of active participation in life by 4.0%, and the score of life satisfaction by 0.088 (and the effects estimated by the three matching methods were similar).

### 3.5. Heterogeneity Analysis

We then grouped the samples for heterogeneity analysis according to urban/rural areas, age groups, whether disabled or not, and the different SES groups. Table 5 shows that the use of home and community care improved health in all dimensions among rural older adults, furthermore it exerted a higher positive influence on the level of social adaptation in urban older adults. Using home and community care significantly promoted the multidimensional health of adults aged 60 to 79 years but had no impact among people aged ≥ 80 years. The use of home and community care significantly improved all health indicators in non-disabled older people; whereas it only significantly improved the levels of cognitive function and life satisfaction for disabled older people. Using this form of care significantly improved all health indicators among those with low SES; whereas it only improved the levels of cognitive function, active participation in life, and life satisfaction in those with high SES.

### 3.6. Robustness Analysis

In addition, we replaced the indicators of multidimensional health to further check the robustness. We utilized the self-reported health status, and retransformed the indicators of depression, cognitive function, and social activities, for the robustness analysis. The self-rated health status of “very good” was scored as 5 and the status of “very bad” was scored as 1. If the score of depression is greater than 10, the binary outcome variable of depression was defined as 1, representing older adults with depression, otherwise, it was scored as 0 [16]. If the score of cognitive function was greater than 24, the binary outcome variable of cognitive function was defined as 1, meaning good cognitive function, otherwise, it was scored as 0 [22]. The variable of social interaction was measured by the four social activities: “visiting, communicating with friends”, “playing mahjong, playing chess, playing cards, going to the community activity room”, “dancing, fitness, qigong practice, etc.”, and “participating in community organization activities”. The results showed that after replacing the outcome variables, the use of home and community care still had a significantly positive impact on the multidimensional health of older people (Table 6).

## 4. Discussion

This is one of the very first studies systematically exploring the effect of using home and community care on the multidimensional health of older people. Based on a national representative sample, we found that using home and community care has a significant positive impact on the multidimensional health of older people. Our results were similar to previous studies which relied on small samples and basic statistical methods showing that the use of home and community care increased the daily activity, ability, and mental health of older people in Japan [11], and that home and community care services maintained the independence of older adults living alone in Korea [23]. We also found that the proportion of those using home and community care was relatively low in China. Therefore, considering the positive effects identified in this study, we suggest that the government should increase the supply of home and community care services and promote older adults’ understanding of such services, thus enhancing their multidimensional health.

We then identified that the use of home and community care had a significantly positive effect on all health indicators in rural older adults and a more pronounced positive effect on the social adaptation of urban older adults. Generally speaking, as most home and community care services in China are in the early stage of development, they belong to the basic daily care and medical services type. Furthermore, in this country, these services could not be as mature as those developed decades ago [6]. Specifically, home and community care in urban China started earlier than in rural areas and have a relatively higher level of development, usually equipped with recreational facilities. In rural China, older adults have a stronger demand for home and community care services. However, such services only commenced operations in recent years [24]. Therefore, once home and community care is supplied to rural areas, it will play a highly prominent role in maintaining self-perceived physical health and bring psychological comfort to rural older people. Considering that rural older adults enjoy much fewer social and economic benefits than their urban counterparts [25], we suggest more support is given towards developing home and community care services in rural areas, thus enhancing their overall well-being.

We also uncovered that the use of home and community care significantly improved all health indicators for those with low SES, but it only had a partially positive effect on the multidimensional health of those with high SES. These results were in line with the heterogeneity results relating to urban/rural analysis, which showed that rural respondents gain greater benefits from the use of home and community care services than their urban counterparts. The Chinese government usually purchases or provides basic home and community care for community residents. Thus, such basic services are usually for free or at low cost [26]. Similar to those from rural areas, those with low SES have high unmet needs for long-term care [27]. Once home and community care services are provided to them, their multidimensional health significantly improves.

Furthermore, we demonstrated that the use of home and community care had no significant impact on the multidimensional health of people aged ≥80 years and exerted no significant impact on the physical health, depression, and active participation in life of disabled older adults. Older adults aged ≥80 years and those with disability need continuous, intelligent, timely, efficient, and coordinated long-term care [28]. Our results suggest that the current home and community care might not fully fulfill the high needs of frail older people. Therefore, we suggest the government improve the quality of home and community services targeted at frail older people, and promote the integrated care model of home and community services and traditional institutional services for the population. This study used the PSM method to correct the endogeneity problem. As for the methodology, this study still had limitations. Firstly, the PSM method cannot account for the bias of unobserved covariates. We relied on the replacement of outcome variables in the robustness analysis showing similar results to the main PSM analysis. Secondly, we only utilized cross-sectional data in this study. Thus, we cannot investigate the time-series effect of using home and community care. Future studies using time-series data are required in this regard. Thirdly, due to data limitations, this study used self-perceived physical health to measure changes in the physical health of older adults and the self-reported use of services to measure the use of home and community care. Future studies are needed using objective physical health or objective use of care indicators. Finally, we observed some relatively small but positive effects on some health indicators due to the influence of using home and community care. Considering that such care is in its early development in China, we believe that, as this form of care develops, its effect on the multidimensional health of older adults will gradually grow.

## 5. Conclusions

Currently, home and community care services are developing quickly in LMICs. We found that approximately 20% of the respondents had used home and community care services. We also found that the use of home and community care services significantly improved the multidimensional health of older adults in China and that the magnitude of the effect differed hugely between urban and rural areas, different age groups, having disabilities or not, and different SES groups. We suggest the government make a special effort to help older adults: in rural areas, with higher ages, and with disabilities, when further improving the quality of home and community care services. In addition, this paper greatly enriches theories involving long-term care as it systematically evaluated the empirical relationship between the use of home and community care services and multidimensional health.

## Figures and Tables

**Figure 1 ijerph-19-15402-f001:**
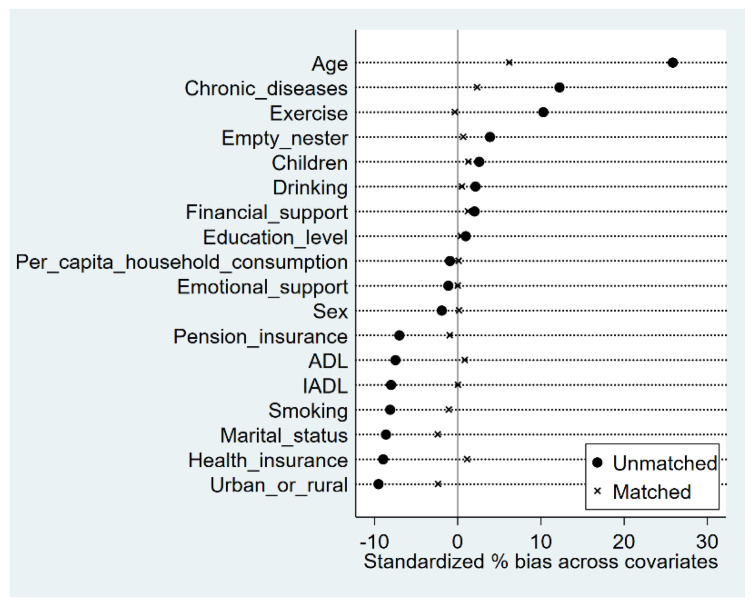
Standardized % bias across covariates standardized differences (%) (nuclear matching).

**Figure 2 ijerph-19-15402-f002:**
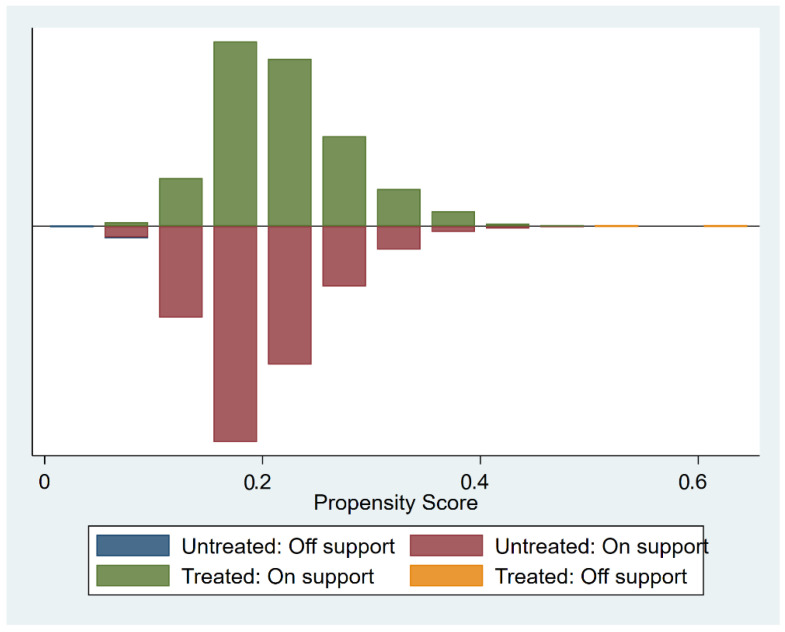
Propensity scores in treatment and control groups (nuclear matching).

**Table 1 ijerph-19-15402-t001:** Variables and their measurement.

	Variable	Measurement	Mean or %
Outcome variables	Change in physical health	1 = Maintenance or improvement, 0 = Deterioration	0.474
	Depression	0~30	8.596
	Cognitive function	0~30	21.176
	Active participation in life	1 = Yes, 0 = No	0.125
	Life satisfaction	1~5	3.280
Explanatory variable	The use of home and community care	1 = Yes, 0 = No	0.202
Covariates	Age	Continuous variable	68.807
	Sex	1 = Male, 0 = Female	0.492
	Urban/rural areas	1 = Rural, 0 = Urban	0.758
	Marital status	1 = Married, 0 = Not married	0.791
	Education level	1 = Illiterate	29.78%
		2 = Primary school	45.01%
		3 = Junior school	15.70%
		4 = High school	8.09%
		5 = Bachelor or above	1.42%
	Being an empty nester	1 = Yes, 0 = No	0.480
	Pension insurance	1 = Pension for public servants and public institution employees	20.28%
		2 = Basic pension for enterprise employees	66.44%
		3 = Pension for land-expropriated farmers, commercial pension insurance	2.55%
		4 = No pension insurance	10.73%
	Health insurance	1 = Urban employee medical insurance or urban and rural resident health insurance	94.53%
		2 = Other health insurance	2.47%
		3 = No health insurance	3.00%
	The number of chronic diseases	0 = 0	52.50%
		1 = 1	28.67%
		2 = 2 and above	18.83%
	ADL	0~18	0.697
	IADL	0~18	1.592
	Smoking	1 = Yes, 0 = No	0.289
	Drinking	1 = Yes, 0 = No	0.349
	Exercise	1 = Yes, 0 = No	0.885
	SES	1 = lowest 25%, 2 = lower25%, 3 = higher 25%, 4 = highest 25%	25%
	The number of children	Continuous variable	2.997
	Financial support from children	Continuous variable	5 244.528
	Emotional support from children	1 = Living with their children or contacting them almost every day	59.62%
		2 = Contact children two or three times a week	29.02%
		3 = Contact children every half month or one month	9.56%
		4 = Other	1.80%

Notes: ADLs = Activities of Daily Living; IADLs = Instrumental Activities of Daily Living; SES = socio-economic status.

**Table 2 ijerph-19-15402-t002:** Estimates of the use of home and community care services.

Variables	Coef.	S.E.	*p*-Value	[95% CI]	
Age	0.049 ***	0.005	<0.001	0.040	0.058
Sex	−0.067	0.073	0.358	−0.211	0.076
Urban/rural areas	−0.076	0.071	0.287	−0.215	0.064
Marital status	−0.063	0.066	0.338	−0.193	0.066
Education level	−0.007	0.032	0.829	−0.071	0.057
Being an empty nester	0.104	0.058	0.071	−0.009	0.218
The number of chronic diseases	0.172 ***	0.033	<0.001	0.107	0.236
ADL	−0.030	0.021	0.144	−0.070	0.010
IADL	−0.032 **	0.011	0.005	−0.054	−0.009
Smoking	−0.120	0.068	0.077	−0.252	0.013
Drinking	0.064	0.064	0.319	−0.062	0.189
Exercise	0.343 ***	0.091	<0.001	0.164	0.523
SES	−0.029	0.025	0.240	−0.078	0.020
The number of children	−0.055 **	0.020	0.007	−0.095	−0.015
Financial support from children	1.24 × 10^−6^	2.19 × 10^−6^	0.571	−3.05 × 10^−6^	5.53 × 10^−6^
Emotional support from children	−0.045	0.039	0.256	−0.121	0.032
Pension insurance	−0.047	0.036	0.191	−0.117	0.023
Health insurance	−0.333 ***	0.082	<0.001	−0.494	−0.172
Constant	−4.267 ***	0.382	<0.001	−5.017	−3.518

Note: *** *p* < 0.01, ** *p* < 0.05; SES = socio-economic status.

**Table 3 ijerph-19-15402-t003:** Results of balance test (nuclear matching).

Variables		Mean		Standardized Differences (%)	*t*-Test	
		Treatment Group	Control Group		*t* Value	*p* > T
Age	Before matching	70.149	68.467		10.11	0.00
	After matching	70.110	69.708	76.1	1.90	0.06
Sex	Before matching	0.484	0.494		−0.76	0.45
	After matching	0.485	0.484	92.4	0.05	0.96
Urban/rural areas	Before matching	0.725	0.766		−3.82	0.00
	After matching	0.724	0.735	75.0	−0.73	0.47
Marital status	Before matching	0.762	0.798		−3.47	0.00
	After matching	0.764	0.774	72.0	−0.74	0.46
Education level	Before matching	2.071	2.062		0.38	0.71
	After matching	2.071	2.068	61.8	0.12	0.91
Being an empty nester	Before matching	0.495	0.476		1.53	0.13
	After matching	0.495	0.492	83.3	0.20	0.84
The number of chronic diseases	Before matching	0.739	0.644		4.87	0.00
	After matching	0.739	0.721	81.0	0.72	0.47
ADL	Before matching	0.597	0.723		−2.83	0.01
	After matching	0.598	0.584	88.9	0.29	0.77
IADL	Before matching	1.397	1.641		−3.07	0.00
	After matching	1.399	1.398	99.8	0.01	1.00
Smoking	Before matching	0.260	0.296		−3.17	0.00
	After matching	0.260	0.265	86.7	−0.34	0.73
Drinking	Before matching	0.357	0.347		0.84	0.40
	After matching	0.357	0.354	76.5	0.16	0.88
Exercise	Before matching	0.910	0.878		3.91	0.00
	After matching	0.910	0.911	96.6	−0.12	0.91
SES	Before matching	2.489	2.499		−0.37	0.71
	After matching	2.491	2.490	89.5	0.03	0.98
The number of children	Before matching	3.028	2.990		1.03	0.30
	After matching	3.031	3.012	51.1	0.40	0.69
Financial support from children	Before matching	5426.800	5198.400		0.79	0.43
	After matching	5433.600	5296.300	39.9	0.39	0.70
Emotional support from children	Before matching	1.529	1.537		−0.44	0.66
	After matching	1.528	1.528	99.9	0.00	1.00
Pension insurance	Before matching	1.992	2.049		−2.78	0.01
	After matching	1.992	2.000	86.2	−0.30	0.76
Health insurance	Before matching	1.060	1.091		−3.33	0.00
	After matching	1.060	1.056	87.6	0.40	0.69

Notes: ADLs = Activities of Daily Living; IADLs = Instrumental Activities of Daily Living.

**Table 4 ijerph-19-15402-t004:** ATT of using home and community care services on the multidimensional health of older adults.

Health Indicators	Matching Methods	ATT	S.E.	*t*-Value	*p*-Value
Change in physical health	K-nearest neighbor matching	0.029 **	0.013	2.26	0.016
	Nuclear matching	0.035 **	0.014	2.50	0.044
	Radius matching	0.030 **	0.013	2.36	0.012
Depression	K-nearest neighbor matching	−0.479 ***	0.165	−2.90	0.001
	Nuclear matching	−0.539 **	0.184	−2.93	0.017
	Radius matching	−0.457 ***	0.166	−2.75	0.002
Cognitive function	K-nearest neighbor matching	0.694 ***	0.126	5.49	<0.001
	Nuclear matching	0.723 ***	0.140	5.16	<0.001
	Radius matching	0.717 ***	0.127	5.64	<0.001
Active participation in life	K-nearest neighbor matching	0.040 ***	0.009	4.46	<0.001
	Nuclear matching	0.041 ***	0.010	4.11	0.001
	Radius matching	0.041 ***	0.009	4.50	<0.001
Life satisfaction	K-nearest neighbor matching	0.088 ***	0.020	4.51	<0.001
	Nuclear matching	0.078 ***	0.022	3.60	0.001
	Radius matching	0.089 ***	0.020	4.51	<0.001

Notes: 500 bootstrapped samples were applied to estimate standard errors. *** *p* < 0.01, ** *p* < 0.05. ATT = average treatment effect

**Table 5 ijerph-19-15402-t005:** Heterogeneity analysis (nuclear matching).

	Change in Physical Health	Depression	Cognitive Function	Active Participation in Life	Life Satisfaction
Urban older adults (*n* = 2349)	0.012	−0.344	0.487 ***	0.076 ***	0.111 ***
Rural older adults (*n* = 7343)	0.033 **	−0.474 **	0.662 ***	0.025 **	0.079 ***
People aged 60 to 79 years (*n* = 8896)	0.029 **	−0.632 ***	0.728 ***	0.042 ***	0.092 ***
People aged ≥ 80 years (*n* = 796)	0.030	−0.784	0.514	0.034	0.033
Non-disabled older adults (*n* = 7350)	0.032 **	−0.494 ***	0.704 ***	0.047 ***	0.055 **
Disabled older adults (*n* = 2342)	0.030	−0.566	0.700 ***	0.023	0.209 ***
SES-high (*n* = 4846)	0.020	−0.345	0.696 ***	0.060 ***	0.071 **
SES-low (*n* = 4846)	0.029 ***	−0.479 ***	0.694 ***	0.040 ***	0.088 ***

Note: *** *p* < 0.01, ** *p* < 0.05.

**Table 6 ijerph-19-15402-t006:** Robustness analysis II (estimated by replacing outcome variables, nuclear matching).

	ATT	S.E.	*t*-Value	*p*-Value
Self-reported health	0.039 **	0.026	1.50	0.044
Depression	0.022 **	0.012	1.86	0.048
Cognitive function	0.042 ***	0.011	3.57	<0.001
Social activities	0.142 ***	0.022	6.59	<0.001

Notes: *** *p* < 0.01, ** *p* < 0.05. ATT = average treatment effect.

## Data Availability

The data used in this study can be found here: https://charls.charlsdata.com/pages/data/111/zh-cn.html (accessed on 7 July 2022).

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
