# Peer review of "Effect of the Use of Home and Community Care Services on the Multidimensional Health of Older Adults"

_ijerph, 2022, doi:10.3390/ijerph192215402_

Round 1

Reviewer 1 Report

Major points:

1.       The introduction needs to be improved. There are no clear transitions, and it is not clear why authors selected “home and community care service” for their study. Is the home and community care service the only way for older adults?

2.       For the entire manuscript, only section 3.4 and 3.5 show some analysis related to home and community care service, other results are based on the whole respondents. So I think the manuscript should be more focus on the subject as the title that authors made. Or provide deeper analysis or other supportive data for this title.

3.       Line 115, the numbers show that the percentage of people having “home and community care service” is low. The following analysis is based on the whole respondents, then what is the meaning to have these home and community care service data in this study?

Minor points:

1.       Table 1, I suggest having some solid/dash lines to separate some rows, otherwise it is not easy to follow and looks messy.

2.       Table 3, how did authors define the control group?

3.       Line 152-155, authors found home and community care service make significant differences based on statistical analysis. But would the differences, such as 0.471 or 0.088, make really differences in practical life?

4.       Table 5, authors analyzed the people having care services based on groups, what is the sample size of each group?

Author Response

  1. The introduction needs to be improved. There are no clear transitions, and it is not clear why authors selected “home and community care service” for their study. Is the home and community care service the only way for older adults?

Response: we have added the transition sentences in the introduction. Now they read: “Currently, two types of long-term care services are mainly available for old adults: institutional care and home and community care. Compared with institutional care, the advantage of home and community care services is that it can not only provide diversified services for older people at a relatively reasonable price but also meet their emotional needs not to leave home.”

  1. For the entire manuscript, only section 3.4 and 3.5 show some analysis related to home and community care service, other results are based on the whole respondents. So I think the manuscript should be more focus on the subject as the title that authors made. Or provide deeper analysis or other supportive data for this title.

Response: We have rewritten the sentences in lines 127-139, making it clear why we used the analytical method. PSM requires that we need to include both users (the treatment group) and nonusers (the control group). Based on the results of matching, we estimated the average treatment effect of using home and community care services on the health of older adults.

  1. Line 115, the numbers show that the percentage of people having “home and community care service” is low. The following analysis is based on the whole respondents, then what is the meaning to have these home and community care service data in this study?

Response: We have added more explanations on why we used the methods in lines 127-129. By including both users and nonusers, we estimated the average treatment effect of using home and community care services on the health of older adults.

Minor points:

  1. Table 1, I suggest having some solid/dash lines to separate some rows, otherwise it is not easy to follow and looks messy.

Response: We have revised Table 1 as suggested.

  1. Table 3, how did authors define the control group?

Response: We rewrote the sentence in line 135-137. And now it reads: “We utilized kernel matching, k-nearest neighbor matching (k=4), and radius matching to match treatment and control groups.” As we wrote, those who did not use home and community care services were defined as the control group.

  1. Line 152-155, authors found home and community care service make significant differences based on statistical analysis. But would the differences, such as 0.471 or 0.088, make really differences in practical life?

Response: we have added another point of limitation. Now it reads: “We observed some positive but relatively small effects for some health indicators due to the influence of using home and community care. Considering that such care is in its early development in China, we believe that as the further development of this form of care, its effect on the multidimensional health of older adults will gradually grow”.

  1. Table 5, authors analyzed the people having care services based on groups, what is the sample size of each group?

Response: We have added the sample sizes of each group in Table 5.

Reviewer 2 Report

Dear Authors minor errors that should be corrected before publication.

Details are in the attachment.

Author Response

Reviewer 2

In general, there are several errors that should be corrected before publication:

Abstract: there is lack of 1-2 sentences of background.

Response: We have added one sentence in the background in the abstract.

Keywords are too general they should be more specific.

Response: We have changed the keywords to make them more specific.

The aim of the study: please be consisted. In abstract, line 8-9, the aim of the study is: This study aimed at estimating the effect of using home and community care services on the multidimensional health of older adults. In the main document, line 49-59: This study is going to filling this gap by systematically assessing the effect of using home and community care on the multidimensional health and exploring the heterogeneity of such effect among older adults in China. Please decide if the purpose of the study was to evaluate the effect of using home and community care services on the multidimensional health of older adults or to fill the research gap in this regard.

Response: We have rephrased the sentence in line 49-59 to make them clear.

In the introduction, it is worth providing some statistics (data on the number of people aged over 60 I your country, the number of social welfare homes, the number of people staying in social welfare homes, who and under what conditions can benefit from care in a social welfare home).

Response: We have added the related information in line 42-47.

Line 38 - provide the source Material and method: please explain on what basis the authors had access to the data, are they generally available, obtained consent?

Response: We have added such information in line 76-77. The data of CHARLS can be obtained via an online application.

Line 91: Please explain abbreviations when first used (ADL, IADL).

Response: We have added the explanations of ADL and IADL in line 111-112.

Results: All abbreviations used in the table should be explained in the footer below the table. Response: We have added the explanations of all abbreviations in the related tables.

Discussion: The authors only once referred to the situation in Japan, what about other countries? It is worth to include information about the results of studies in other countries in this area.

Response: We have added the information of the results in Korea in line 254-255.

The conclusion, too short, should clarify the main contribution of the paper and the value added to the field

Response: We have added more information in the conclusion.

Reviewer 3 Report

Congratulations to the authors on a well-written and topic article that has a great deal of opportunity to inform. The article overall is well written,  I feel it could be better to elaborate more in the introduction section to increase readability though. Some individual comments on the introduction below:

1. The introduction is really well written, and I enjoyed and learned from it but its a little bit short. I also feel that it needs to highlight more specific knowledge gaps that your study will fill –. 

Author Response

Reviewer 3

Congratulations to the authors on a well-written and topic article that has a great deal of opportunity to inform. The article overall is well written, I feel it could be better to elaborate more in the introduction section to increase readability though. Some individual comments on the introduction below:

  1. The introduction is really well written, and I enjoyed and learned from it but its a little bit short. I also feel that it needs to highlight more specific knowledge gaps that your study will fill –.

Response: We have added quite some sentences in the introduction.

Reviewer 4 Report

This paper contributes to the broad literature on the influence of home and community care services on the health outcomes of older adults, focusing on the LMICs, especially in China. It does so by using the China Health and Retirement Longitudinal Study (2018) and mainly relying on the propensity score matching method. The topic, the setting, and the method all militate in favor of publication. On the other hand, key issues about the paper’s theoretical foundation and its operationalization warrant caution.

As for the introduction, I suggest the authors review the existing literature on the relationship between home/community care services and health among older adults. Why this form of care is especially beneficial to older people’s health. What we have known and what we still don’t know.

The authors should indicate the year of the survey data used for this study in the data section. For the measure of physical health. I am inclined to regard it as self-rated health. The CHARLS has a scale to measure the functional health of older adults. I would suggest the author consider using the functional health scale to measure physical health rather than claiming self-rated health as physical health.

The measure of community/home care is somewhat arbitrary. Could you cite any related literature to support your operationalization? An important covariant that should be included is health behaviors, such as smoking, drinking, or exercising. The health behaviors should be included in the logit model to predict p-scores.

Regarding the results, why is the interpretation of physical health in percentage? My understanding is that physical health is a binary variable. How was PSM applied for a non-continuous variable?

For the heterogeneity analysis, I wonder if the author could split the sample by their SES. I suspect those older adults who can afford community/home care are more likely to be in high SES groups. Those privileged older people may already have better health than their poor counterparts.

I'm also concerned about your IVs. Although the IVs have passed statistical tests, they do not make intuitive sense. The number of children and care from relatives or friends are definitely correlated with not only home and community care services (x) but also health outcomes (y). I understand the authors tried so hard to exclude the possibility of endogeneity. But bad IVs may undermine the reliability of your results.  

Author Response

Reviewer 4

This paper contributes to the broad literature on the influence of home and community care services on the health outcomes of older adults, focusing on the LMICs, especially in China. It does so by using the China Health and Retirement Longitudinal Study (2018) and mainly relying on the propensity score matching method. The topic, the setting, and the method all militate in favor of publication. On the other hand, key issues about the paper’s theoretical foundation and its operationalization warrant caution.

As for the introduction, I suggest the authors review the existing literature on the relationship between home/community care services and health among older adults. Why this form of care is especially beneficial to older people’s health. What we have known and what we still don’t know.

Response: We have added information on why this form of care is beneficial to older people’s health from a theoretical point of view. And this theoretical point is the basis of our hypothesis and the whole analysis.

The authors should indicate the year of the survey data used for this study in the data section. For the measure of physical health. I am inclined to regard it as self-rated health. The CHARLS has a scale to measure the functional health of older adults. I would suggest the author consider using the functional health scale to measure physical health rather than claiming self-rated health as physical health.

Response: The functional health, such as ADLs, IADLs were used as covariates in this study since they affect the matching very much, deciding whether adults people to use home and community care itself. As we observe, more severely disabled older adults use institutionalized care. More importantly, considering that many diseases of old age are degenerative diseases, the services themselves cannot improve ADLs or IADLs, but can only prevent the physical health from getting worse. So the basic logic in this study was to look at whether home and community care services can maintain or improve the physical health of older adults, so we chose this variable as the outcome measure of physical health.

The measure of community/home care is somewhat arbitrary. Could you cite any related literature to support your operationalization? An important covariant that should be included is health behaviors, such as smoking, drinking, or exercising. The health behaviors should be included in the logit model to predict p-scores.

Response: We have added smoking, drinking, or exercising as covariates and redone all the related analysis. In addition, the measure of home and community care was totally based on the questionnaire. We have added the related sentence in the limitation part. It belonged to the self-reported use of home and community care.

Regarding the results, why is the interpretation of physical health in percentage? My understanding is that physical health is a binary variable. How was PSM applied for a non-continuous variable?

Response: The physical health is a binary variable. PSM can be applied for a binary outcome.

For the heterogeneity analysis, I wonder if the author could split the sample by their SES. I suspect those older adults who can afford community/home care are more likely to be in high SES groups. Those privileged older people may already have better health than their poor counterparts.

Response: We have split the sample by their SES to do heterogeneity analysis. We have added the related contents in Table 5, results (line 203-206), and discussion (line 276-285).

I'm also concerned about your IVs. Although the IVs have passed statistical tests, they do not make intuitive sense. The number of children and care from relatives or friends are definitely correlated with not only home and community care services (x) but also health outcomes (y). I understand the authors tried so hard to exclude the possibility of endogeneity. But bad IVs may undermine the reliability of your results.  

Response: After adding smoking, drinking, or exercising as covariates, the IV models still held. And the model passed all the tests. As we indicated, two other studies have used these two IVs as well. However, after considering the comments of the reviewer, we decided to delete this part. By replacing the indicators of the multidimensional health, our model still had similar results, showing that the model passed the robustness check.

Round 2

Reviewer 1 Report

The manuscript is improved but the improvement is not enough. For example, authors said the sample size is added into Table 5, but I don't see sample size in the revised version. Can the authors please take reviewer comments seriously?

Author Response

The manuscript is improved but the improvement is not enough. For example, authors said the sample size is added into Table 5, but I don't see sample size in the revised version. Can the authors please take reviewer comments seriously?

Response: We changed a lot between co-authors in our first round of review. The sample size was added in another version. We are so sorry for that. Now the sample size was added in Table 5.

Reviewer 4 Report

The revised version of this paper addresses most of the concerns I raised in the previous review round, and I appreciate the authors’ efforts to improve the manuscript.

I have only one remaining comment. It seems the authors did not address my first comment from the previous round.

The author claimed they have added information on why this form of care is beneficial to older people’s health from a theoretical point of view. They did not specify where they addressed this issue in the manuscript. I also didn’t find reviews of the existing literature on the relationship between home/community care services and health outcomes. Is the relationship positive or negative? Does it depend on specific contexts or subgroups?

Author Response

The revised version of this paper addresses most of the concerns I raised in the previous review round, and I appreciate the authors’ efforts to improve the manuscript.

I have only one remaining comment. It seems the authors did not address my first comment from the previous round.

The author claimed they have added information on why this form of care is beneficial to older people’s health from a theoretical point of view. They did not specify where they addressed this issue in the manuscript. I also didn’t find reviews of the existing literature on the relationship between home/community care services and health outcomes. Is the relationship positive or negative? Does it depend on specific contexts or subgroups?

Response: we have added where we are going to address this issue in lines 66-68 in the manuscript. In addition, we have rephrased the literature part in lines 60-66, aiming at explicitly reviewing the existing literature on the relationship between home and community care and health outcomes. As we wrote, the existing evidence is very limited. In addition, the existing evidence has limitations in their methodology. So their estimates might have flaws. That is why we did this study systematically exploring the relationship between using home and community care and multidimensional health among older adults in China.